# Chemical characteristics of brown carbon in atmospheric particles at a suburban site near Guangzhou, China

Yi Ming Qin[1,#], Hao Bo Tan[2], Yong Jie Li[3], Zhu Jie Li[2,4], Misha I. Schurman[5], Li Liu[2,6], Cheng Wu[7], and Chak K. Chan[1]

[1]School of Energy and Environment, City University of Hong Kong, Hong Kong, China

[2]Key Laboratory of Regional Numerical Weather Prediction, Institute of Tropical and Marine Meteorology, China Meteorological Administration, Guangzhou, China

[3]Department of Civil and Environmental Engineering, Faculty of Science and Technology, University of Macau, Macau, China

[4]School of Environmental Science and Engineering, Nanjing University of Information Science and Technology, Nanjing, China

[5] Zephyr Research Consultants, USA

[6]Department of Atmospheric Science, Sun yat-sen University, Guangzhou, China

[7]Institute of Mass Spectrometer and Atmospheric Environment, Jinan University, Guangzhou, China

[#] now at School of Engineering and Applied Sciences, Harvard University, Cambridge, MA, USA

Correspondence to: Chak K. Chan (chak.k.chan@cityu.edu.hk) and Yong Jie Li (yongjieli@umac.mo)

Abstract:

Light-absorbing organic carbon (or brown carbon, BrC) in atmospheric particles has received much attention for its potential role in global radiative forcing. While a number of field measurement campaigns have differentiated light absorption by black carbon (BC) and BrC, the chemical characteristics of BrC are not well understood. In this study, we present co-located real-time light absorption and chemical composition measurements of atmospheric particles to explore the relationship between the chemical and optical characteristics of BrC at a suburban site downwind of Guangzhou, China from November to December 2014. BrC and BC contributions to light absorption were estimated using measurements from a seven-wavelength aethalometer, while the chemical composition of non-refractory $PM_1$ was measured with a high resolution time-of-flight aerosol mass spectrometer (HR-ToF-AMS). Using the Absorption Angstrom Exponent (AAE) method, we estimated that BrC contributed 23.6% to the total aerosol absorption at 370 nm, 18.1% at 470 nm, 10.7% at 520 nm, 10.7% at 590 nm, and 10.5% at 660 nm. Biomass burning organic aerosol (BBOA) has the highest mass absorption coefficient among sources of organic aerosols. Its contribution to total brown carbon absorption coefficient decreased but that of low-volatility oxygenated organic aerosol (LVOOA) increased with increasing wavelength, suggesting the need for wavelength-dependent light absorption analysis for BrC in association with its chemical makeup. Clear correlations of N-containing ion fragments with absorption coefficient were observed. These correlations also depended on their degrees of unsaturation/cyclization and oxygenation. While the current study relates light absorption by BrC to ion fragments, more detailed chemical characterization is warranted to constrain this relationship.

## 3 Introduction:

Atmospheric particles participate considerably in the global climate direct effect via their light-scattering (e.g., sulfate) and/or light-absorbing components (e.g., black carbon, BC). BC is major contributor to light absorption that leads to positive radiative forcing, increasing the average temperature of the atmosphere. The BrC absorption contribution to total aerosol light absorption can reach 20–50% over regions dominated by seasonal biomass burning and biofuel combustion (Feng et al., 2013).A significant difference in optical feature of BrC and BC is that BrC absorbs light primarily at UV and short-visible wavelengths with the absorption decreasing significantly at long wavelengths, while BC absorbs strongly and constantly throughout the UV to visible spectrum(Andreae and Gelencsér, 2006; Bergstrom et al., 2007; Bond and Bergstrom, 2006). In global climate models, the direct radiative forcing of organic aerosols at the top of atmosphere can shift from cooling ($-0.08$ Wm$^{-2}$) to warming ($+0.025$ Wm$^{-2}$) when strong BrC absorption is included(Feng et al., 2013). However, uncertainties in the sources, formation, chemical composition, and absorption properties of BrC hinder more accurate estimations of radiative forcing induced by atmospheric particles.

BrC is operationally defined and has many chemical constituents, which makes chemical characterization quite challenging. Both primary and secondary organic aerosols can act as BrC (Laskin et al., 2015). For example, biomass burning organic aerosol (BBOA) has been identified as a contributor to BrC in rural areas in the southern United States , while coal combustion organic aerosol (CCOA) contributes substantially to BrC during winter in Beijing (Yan et al., 2017). Species from secondary formation processes, such as humic-like substances (HULIS) formed by in-cloud processing (Rinco et al., 2009), species from gas-phase photo-oxidation of volatile organic compounds (VOCs) in the presence of NO$_x$, and species from reactions between carbonyl compounds and ammonia in the aqueous film at the particle surface, can also contribute to BrC (Gen et al., 2018; Laskin et al., 2010; Liu et al., 2015). Highly conjugated organics, nitro-aromatic compounds, imidazoles, and other N-heterocyclic compounds have been found in BrC (Laskin et al., 2015; Lin et al., 2016). Sun et al. (2007) also found that light-absorbing organic molecules in BrC are likely large (i.e., possessing > 18 carbon atoms); these molecules are generally highly unsaturated and contain three or more oxygen atoms and/or one or more nitrogen atoms.

The Pearl River Delta (PRD) region, one of the most economically developed regions in China, suffers under air pollution from a variety of sources (Chan and Yao, 2008; Li et al., 2017). Source

apportionment using positive matrix factorization (PMF) analysis of mass spectral data sets from
high resolution time-of-flight aerosol mass spectrometry (HR-ToF-AMS) has revealed that the
organic aerosol (OA) in this region arises from traffic emissions (i.e., hydrocarbon-like organic
aerosol, or HOA), biomass burning (BBOA), cooking (COA), and secondary formation (i.e.,
oxygenated organic aerosols, or OOAs). In the PRD, HOA is often the largest contributor to OA
at urban sites (He et al., 2011), while SOA plays a more important role at rural sites (Gong et al.,
2012; Huang et al., 2011). BBOA has also been found to contribute significantly to total OA in the
PRD region, with contributions of 24% at an urban site in Shenzhen (He et al., 2011) and 14% and
25% at rural sites in Heshan and Kaiping, respectively (Gong et al., 2012; Huang et al., 2011).
Yuan et al. (2016) attributed 6-12% of the total aerosol absorption at 405 nm at a rural site in the
PRD to BrC; the authors found higher BrC contributions during fall, which they ascribed to
biomass burning (BBOA) activities nearby. However, the BrC components responsible for light
absorption remain relatively unknown; this hinders a thorough understanding of the relationships
between optical properties and chemical characteristics and, in turn, the realization of a generalized
framework that can be extended to other sources and regions.
In this work, we present simultaneous measurements of aerosol chemical composition and light
absorption of aerosol particles at a suburban site downwind of Guangzhou in the PRD, China.
Contributions of BC and BrC to total aerosol light absorption were differentiated and quantified
using measurements from a seven-wavelength aethalometer. Sources of OA, which were
determined using PMF analysis, were correlated to BrC light absorption to identify the major
contributor(s) to short-wavelength light absorption. More detailed chemical characteristics, such
as N-containing ion fragments, the degree of unsaturation (indicated by the ion double bond
equivalent, or ion DBE), and the degree of oxygenation (indicated by the number of oxygen atoms
in ions), were also used to investigate the structural characteristics of BrC related to light
absorption.
Methodology
1.  Sampling site
We conducted field measurements at the Guangzhou Panyu Atmospheric Composition Station
(GPACS, 23°00′ N, 113°21′ E), on the periphery of Guangzhou, China, from November 7, 2014
to January 3, 2015. The GPACS is located on top of a hill with an altitude of approximately 150
m a.s.l. (Cheung et al., 2016; Tan et al., 2013; Zou et al., 2015); it is approximately 15 km south
of the city center and was downwind of the central city throughout the sampling period, during
which north winds prevailed (Qin et al., 2017).
2. Measurements and data analysis
Aerosol light absorption was measured with a seven-wavelength aethalometer (Magee Scientific,
model AE33) at 370, 470, 520, 590, 660, 880, and 950 nm. Ambient air was drawn through a 2.5-
μm cut-off inlet at 2 L/min before entering the aethalometer; particles were collected on the filter
substrate, and light attenuation at the wavelengths above was recorded continuously. A diffusion
drier was used to dry the sampled air stream, which reduced the RH of the air to below 30 %. The
optical properties of the collected particles were determined by comparing light attenuation in
particle-laden and particle-free filter areas (Weingartner et al., 2003). To convert aerosol particles
light attenuation coefficients at the filter substrate to the light absorption coefficients suspended in
the air, a real-time compensation parameter $k$ and a fixed multiple scattering parameter $C$ were
used. The real-time loading effect correction was performed using two parallel measurements of
optical attenuations at different accumulation rates. $C_{ref}$ =2.14 for quartz filter and $C_{ref}$=1.57 for
tetrafluoroethylene (TFE)-coated glass filter were recommended from previously studies for the
fresh soot particles (Drinovec et al., 2015; Weingartner et al., 2003). However, with the presence
of semi-volatile oxidation products, significantly higher values ($C$ =3.6±0.6) were observed in the
organic coating experiment using a quartz filter (Weingartner et al., 2003). Wavelength
dependence of C has also been reported in the literature (Arnott et al., 2005; Schmid et al., 2006;
Segura et al., 2014) . A broad range of C (from 2.8 to 7.8) at several sites was also used by Collaud
Coen et al. (2010). As the multiple scattering parameter (C) may be site specific, we further
compared the absorption from AE33 with cavity ring-down spectroscopy (CRD, Hexin XG-1000)
and Nephelometer (TSI, 3563). The nephelometer was calibrated by $CO_2$ weekly during the field
campaign. Particle-free air was checked once a day. The CRD was calibrated using polystyrene
spheres with known indices of refraction before the campaign. We extracted the light absorption
based on extinction and scattering measurements from cavity ring-down spectroscopy and
Nephelometer, respectively, as below.
$$b_{abs} = \sigma_{ext} - \sigma_{sp} \qquad (1)$$

where $b_{abs}, \sigma_{ext}$ and $\sigma_{sp}$ are absorption coefficient, extinction coefficient and scattering
coefficient.
The scatter plot of absorption at 532 nm from measurement from the aethalometer (AE33) and that
calculated from CRD and Nephelometer (CRD-Neph) is displayed in Figure 1. AE 33 absorption
coefficient was higher than the absorption estimated from Eq. 1. by a factor of 2.10. Therefore, the
final multiple scattering parameter (C) was set to $C_{final}$ = $C_{ref}$ × 2.10 = 3.29. This value is
comparable with previous aethalometer measurements (C=3.48) in the PRD region (Wu et al.,

100  2009, 2013).

The non-refractory chemical composition of submicron aerosols was measured with an Aerodyne
HR-ToF-AMS (Aerodyne Research Inc., Billerica, MA, USA). Briefly, the AMS collected five-
minute-average particle mass spectra for the high-sensitivity V plus particle time-of-flight (PToF)
mode and the high-resolution W mode. AMS data analysis was performed using the SQUIRREL
(v1.56D) and PIKA (v1.15D) toolkits in Igor Pro (WaveMetrics Inc., Lake Oswego, OR). Source
apportionment was performed via PMF analysis with Multilinear Engine 2 (ME-2) via the SoFi
interface (Canonaco et al., 2013). Five factors, including HOA, COA, BBOA, semi-volatile
oxygenated organic aerosol (SVOOA), and low-volatility oxygenated organic aerosol (LVOOA),
were resolved (Qin et al., 2017). The campaign average OA composition was dominated by
surrogates of SOA (SVOOA + LVOOA). However, freshly-emitted hydrocarbon-like organic
aerosols (HOA) contributed up to 40.0% of OA during high-OA periods; during nighttime, HOA
contributed 23.8% to 28.4% on average. BBOA contributed 9.6% (1.87 µg/m$^3$) of total OA in
November and 6.5% (1.38 µg/m$^3$) in December. AMS data treatment was discussed in detail in
Qin et al. (2017). Data from a thermo-optical elemental carbon and organic carbon (ECOC)
analyzer (Sunset Laboratory Inc.) were also used for comparison.
Results and discussion
1.  Aerosol absorption
Figure 2a shows the box-whisker plot of aerosol absorption coefficients ($b_{abs}$) from 370 nm to 950
nm from the aethalometer measurements during the campaign. The campaign-average absorption
coefficients were 56.00 Mm$^{-1}$ at 370 nm, 40.99 Mm$^{-1}$ at 470 nm, 34.76 Mm$^{-1}$ at 520 nm, 29.91
Mm$^{-1}$ at 590 nm, 26.69 Mm$^{-1}$ at 660 nm, 18.06 Mm$^{-1}$ at 880 nm, and 16.71 Mm$^{-1}$ at 950 nm.
In multi-wavelength absorption measurements, the total absorption Ångström exponent (AAE) can
be calculated by a power-law fitting of the absorption coefficient over all available wavelengths.
AAE of unity has been widely used for pure black carbon, while a shift to higher AAE value has
been observed with the presence of brown carbon. The reason behind is that BrC has a much
stronger absorption at UV and short visible wavelengths than at long visible wavelengths, which
yields a steeper curve (Andreae and Gelencsér, 2006; Bergstrom et al., 2007; Bond and Bergstrom,
2006). The presence of non-absorbing OA shells over BC cores may also lead to a shift of AAE
(Gyawali et al., 2009). This latter possibility is analyzed in a separated manuscript (Li et al., in
preparation). Briefly, a Mie theory model was used to estimate the AAE for BC-containing
particles ($AAE_{BC}$) at core-shell scenarios with different refractive indexes. $AAE_{BC}$ is sensitive to
specific refractive index of core and shell of the particles and the size of the particle. The size
distribution is from scanning mobility particle sizer and aerodynamic particle sizer measurement,
and we vary the refractive index of the core and shell in the model. The method is adopted from
Tan et al. (2016) . In general, AAEBC increases as the real part refractive index of the core
increases or the imaginary decreases, or alternatively real part of the shell increases. The $AAE_{BC}$
ranges from 0.67-1.03 across the different scenario (Table S1). As shown in Figure 2b, the AAE
values, which average at 1.43, are almost always higher than 1, indicating appreciable
contributions from BrC to particle light absorption at this site.

To further explore the importance of BrC at this site, BrC absorption at a short wavelength $\lambda_1$
($b_{BrC,\lambda1}$) can be derived by subtracting BC absorption ($b_{BC,\lambda1}$) from the total aerosol absorption
(Lack and Langridge, 2013) via:

$$b_{BrC,\lambda1} = b_{\lambda1} - b_{BC,\lambda1} \qquad (2)$$

where absorption $b_{\lambda1}$ is the measured absorption at the short wavelength $\lambda_1$. BC absorption at $\lambda_1$
($b_{BC,\lambda1}$) can be obtained from the AAE value of BC ($AAE_{BC}$) via:

$$b_{BC,\lambda1} = b_{\lambda2} \times (\lambda_2 / \lambda_1)^{AAE_{BC}} \qquad (3)$$

where $b_{\lambda2}$ is the absorption at a longer wavelength $\lambda_2$ (880 nm), which is assumed to have no
contributions from BrC or dust (Drinovec et al., 2015; Zhu et al., 2017). The uncertainty involved
in attributing BrC and BC absorption at short wavelengths has been explored explicitly by Lack
and Langridge (2013). This uncertainty is primarily from uncertainty of choice of $AAE_{BC}$. Based
on the $AAE_{BC}$ from Mie theory model, a sensitivity analysis of BrC contribution to total light
absorption is presented in Figure S1.
Figure 3 shows the $b_{abs}$ attributed to BC and BrC ($b_{BC}$ and $b_{BrC}$) at different wavelengths. Aerosol
light absorption coefficients were dominated by BC, but $b_{BrC}$ was not negligible, especially at short
wavelengths. The campaign-average $b_{BrC}$ values were 13.67, 7.56, 4.49, 3.22, and 2.81 $Mm^{-1}$ at
370, 470, 520, 590, and 660 nm, respectively; BrC absorption contributed 23.6%, 18.1%, 10.7%,
10.7%, and 10.5% of the total absorption at the corresponding wavelengths. The proportions of
BrC and BC in our campaign were slightly higher than those reported an earlier study in the PRD
by Yuan et al. (2016). In their study, the average light absorption contributions of BrC during
Shenzhen winter, Shenzhen fall, and Heshan fall campaigns were 11.7%, 6.3%, and 12.1% at 405
nm and 10.0%, 4.1%, and 5.5% at 532 nm, respectively.
Figure 4 shows the diurnal variations of both $b_{BrC}$ and $b_{BC}$ at 370, 470, 520, 590, and 660 nm,
respectively. In general, the diurnal cycles of $b_{BrC}$ and $b_{BC}$ share similar patterns, indicating that
they may have similar sources. However, it should be noted that some OA factors, such as BBOA
and HOA, also share similar patterns(Qin et al., 2017). Overall, there were two peaks at each
wavelength. The first peak appeared in the morning at around 8:00 LT, with a peak before 8:00
LT for longer wavelength and after 8:00 LT for shorter wavelength. The second peak appeared at
21:00 LT and its intensity decreased until 24:00 LT. These changes may be attributed to diurnal
changes in BrC sources, which most likely originated from crop residual burning in fall and winter
in nearby regions (Wang et al., 2017). The diurnal variations of the different wavelengths were not
significantly different, although short wavelengths exhibited more obvious diurnal variations.

2. Correlation of light absorption by BrC with OA components
To explore the possible sources of BrC, correlations were determined between $b_{BrC}$ at 370 nm
($b_{BrC,370}$) and various OA types. Data at 370 nm were chosen (over data at longer wavelengths) for
their higher signal-to-noise ratios and larger contributions of BrC to light absorption. Figure 5
shows that BBOA concentrations and $b_{BrC,370}$ were well correlated (Pearson's correlation
coefficient, $R_p = 0.58$). More interestingly, a moderate correlation ($R_p = 0.40$) was also found
between $b_{BrC,370}$ and the LVOOA mass concentration. Although the LVOOA factor was not further
resolved into OOA factors with biomass origins, it is likely that a portion of LVOOA was formed
from biomass burning precursors through either gas-phase oxidation or heterogeneous reactions.
Satish et al. (2017) found correlations between BrC absorption and both primary BBOA and
BBOA-related SVOOA factors. They also reported that the slope of the correlation between
$b_{BrC,370}$ and BBOA (slope = 1.35) was 4.8 times higher than that between $b_{brc,370}$ and one of the
biomass burning SVOOA factors (slope = 0.28), indicating that aging may have reduced the
absorption capacity of biomass-related OA.
Multiple regression analysis was also used to resolve the correlation factors of each OA component
($m^2$ $g^{-1}$) at each wavelength.
$b_{BrC}$= a*[HOA]+b*[COA]+c*[BBOA]+d*[SVOOA]+e*[LVOOA]          (4)
where a, b, c, d, e indicates the correlation factors of each OA component ($m^2$ $g^{-1}$) and [..] indicates
the mass concentration of each OA component. These correlation factors obtained are equivalent
to MAC mass absorption coefficient (MAC) of each OA component. We will use these factors to
compare with MAC reported in the literature later.
Washenfelder et al. (2015) reported a MAC of 1.3 ± 0.06 $m^2$ $g^{-1}$ using the $b_{BrC}$ at 365 nm for
BBOA in the rural southeastern United States, which was 40 to 135 times higher than the MAC
values reported for other OA factors. Di Lorenzo et al. (2017) found that both BBOA and more-
oxidized oxygenated organic aerosol (MO-OOA) were associated with water soluble BrC and that
the MAC of BBOA doubled that of MO-OOA. However, Forrister et al. (2015) observed that BrC
in wildfire plumes had a lifetime of roughly 9 to 15 hours, probably due to conversion to SOA
with lower light absorption capacity. In our study, the MAC (correlation factor in Table 1) of
BBOA at 370 nm was 3.4 ± 0.41 $m^2$ $g^{-1}$, roughly 3.4 times that of LVOOA (1.04 ± 0.08 $m^2$ $g^{-1}$).
Like the studies listed above (Forrister et al., 2015; Di Lorenzo et al., 2017; Washenfelder et al.,
2015), our results suggest that the absorption coefficient of nascent BBOA is higher than that of
its aged counterpart at short wavelength. However, it should be noted that LVOOA might consist
of some other non-absorbing SOA components with no biomass origin. It is therefore important to
consider chromophore lifetimes when modeling light absorption by BrC. As noted in Laskin et al.
(2015), the physicochemical properties of chromophores in BrC may exhibit dynamic changes that
are not yet sufficiently understood. In addition, the difference between MAC values of BBOA and
LVOOA decreased for longer wavelengths. The MAC values of BBOA were roughly 3.4, 1.8, 1.5,
1.48, and 0.80 times those of LVOOA at 370, 470, 520, 570, and 660 nm, respectively.   The
contribution to total absorption coefficient also varied with wavelengths. The contribution from
BBOA decrease from 25.8% to 10.1% from 370 nm to 660 nm, while the contribution from
LVOOA increase from 49.3% to 60.2 % from 370nm to 660nm. The contribution of HOA was
more stable across different wavelengths but was also significant, likely due to the high mass
concentration of HOA. The exponential decay of $b_{abs}$ for different light-absorbing components was
shown in Figure 7. The fitted AAE values for those components are 3.52, 3.28, 5.50 and 2.67 for
total BrC, HOA, BBOA and LVOOA respectively. These results indicate that variability of AAE
values ranging from different sources which is likely inherent to the chemical variability of BrC
constituents. Altogether, these observations indicate that the wavelength-dependent light
absorption of different OAs must be considered in light absorption models.
3.   Correlation of $b_{BrC}$ with N-containing organic ions
The chromophores in BrC that are responsible for OA light absorption are not well characterized.
Structurally, light absorption depends on the extent of $sp^2$ hybridization, in which π electrons are
usually found (Bond and Bergstrom, 2006). Of the elements commonly found in OA, both C and
N have strong tendencies toward $sp^2$ hybridization. It has also been found that, despite their small
OA mass fraction contributions, N-containing organic species in OA can be responsible for
appreciable light absorption (Chen et al., 2016; Laskin et al., 2015). Thus, we examined the
correlations between $b_{BrC}$ and N-containing ions from AMS measurements. These ion fragments,
including the $C_xH_yN^+$ and $C_xH_yO_zN^+$ families, likely originated from N-heterocyclic compounds.
Figure 6 shows that the mass loadings of $C_xH_yN^+$ and $C_xH_yO_zN^{++}$ families are correlated with $b_{BrC}$
at 370 nm and that correlations are stronger for fragments containing both N and O atoms. These
results are consistent with Chen et al. (2016), who suggested that organic compounds with O and
N atoms might contribute substantially to total light absorption and fluorescence in OA
components.
The effects of oxygenation (as indicated by the number of O atoms in an ion) and
unsaturation/cyclization (as indicated by the ion double bond equivalent, or ion DBE) were also
examined for each $C_xH_yN^+$ and $C_xH_yO_zN^+$ ion family. Several studies found that species with high
DBE values may have substantial network of conjugated double bonds and likely contribute to
light absorption (Budisulistiorini et al., 2017; Laskin et al., 2014; Lin et al., 2016). The ion DBE
represents the number of double bonds (unsaturation) or rings (cyclization) that an ion contains
and is calculated on the basis of the elemental formula via the following equation:
$$DBE = C + 1 - H/2 - X/2 + N/2 \qquad (4)$$
where C, H, X, and N are the number of carbon, hydrogen, halogen (Cl, Br, I, and F), and nitrogen
atoms present in the ion, respectively.
Figure 8a shows the correlation coefficients between bBrC at all available wavelengths and the
mass loadings of each ion in $C_xH_yN^+$ and $C_xH_yNO_z^+$ families at different DBE values. For the
$C_xH_yN^+$ family, $R_p$ increased as DBE increased across all wavelength, suggesting that bBrC was
better correlated with fragments with higher degrees of unsaturation or cyclization. And increasing
trend of Rp as DBE increased is more obvious for short wavelengths (e.g. $\lambda$ at 370 nm and 470
nm), suggesting that the absorption at short wavelengths are more associated with the unsaturation
or cyclization. Indeed, in saturated organics, light absorption involves excitation of n electrons,
which requires more energy and, therefore, shorter incident wavelengths (e.g., short UV). In
unsaturated organics, the delocalized $\pi$ electrons are in clusters of sp2 hybrid bonds and in longer
conjugated systems, such that the energy difference between the excited state and the ground
state goes down, which makes the absorption band shift to longer wavelengths. These structural
features may explain in part the increased correlation between mass loadings of the $C_xH_yN^+$ family
and light absorption with decreasing ion saturation. For the $C_xH_yNO_z^+$ family, we did not observe
obvious trends in the correlation coefficient with changing degree of saturation/cyclization (Figure
8b). This phenomenon is consistent across different wavelength. However, the overall Pearson's
Rs of $b_{BrC}$ with $C_xH_yNO_z^+$ were higher than those with $C_xH_yN^+$. The Rp for each group of ions is
higher at short wavelengths ($\lambda$ at 370 nm and 470 nm).

## Conclusions

This paper presents collocated, real-time atmospheric particle light absorption and chemical
composition measurements at a suburban site in PRD, China. While BC dominated aerosol light
absorption, BrC also contributed to absorption at short wavelengths. The aerosol light absorption
coefficients of BrC were 13.67, 7.56, 4.49, 3.22, and 2.81 Mm$^{-1}$ at 370, 470, 520, 590, and 660 nm,
respectively, and BrC contributed 23.6%, 18.1%, 10.7%, 10.7%, and 10.5% of the total absorption
at the corresponding wavelengths. Hydrocarbon-like organic aerosol (HOA), biomass burning
organic aerosol (BBOA) and low-volatility oxygenated organic aerosol (LVOOA) were also
substantial for the source of BrC. At short wavelength (370 nm), the mass absorption coefficient
of BBOA was higher than those of HOA and LVOOA. However, the difference between the mass
absorption coefficients of BBOA and other OA factors decreased with increasing wavelength. The
contribution of different OA sources to total absorption coefficient also varied with wavelengths.
Such a wavelength dependent trend is also observed for their contribution to total BrC absorption
coefficients. $C_xH_yN^+$ and $C_xH_yO_zN^+$, were likely the chromophores responsible for the observed
BrC light absorption. The mass loadings of $C_xH_yN^+$ and $C_xH_yO_zN^+$ ion families became better
correlated with the BrC light absorption coefficient as their degrees of unsaturation/cyclization and
oxygenation increased. This study shows wavelength-dependent light absorption by BrC is
strongly influenced by moderately specific molecular characteristics such as degrees of
unsaturation/ cyclization and oxygenation. An exploration of the absorptive properties of more
specific molecular features, such as the chemical identities of BrC constituents, would require a
more detailed chemical characterization of the highly complex OA composition.

## Acknowledgements

This work was supported by the National Key Project of the Ministry of Science and Technology
of the People's Republic of China (2016YFC0201901, 2016YFC0203305). The authors would
like to acknowledge Hong Kong University of Science and Technology for the use of their AMS.
We also thank Jianhuai Ye for fruitful discussion. Chak K. Chan would like to acknowledge the
Science Technology and Innovation Committee of Shenzhen municipality (project no. 41675117).
Yong Jie Li gratefully acknowledges support from Science and Technology Development Fund of
Macau (FDCT-136/2016/A3).

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

Table
Table 1. Multilinear regression analyses between $b_{BrC}$ at each wavelength and mass loading of different OA factors from AMS-PMF/ME-
451  2.

| | 370 nm | | 470 nm | | 520 nm | | 590 nm | | 660 nm | |
|---|---|---|---|---|---|---|---|---|---|---|
| | Correlation factor $(m^2\,g^{-1})$ | Contribution to $b_{BrC}$ | Correlation factor $(m^2\,g^{-1})$ | Contribution to $b_{BrC}$ | Correlation factor $(m^2\,g^{-1})$ | Contribution to $b_{BrC}$ | Correlation factor $(m^2\,g^{-1})$ | Contribution to $b_{BrC}$ | Correlation factor $(m^2\,g^{-1})$ | Contribution to $b_{BrC}$ |
| HOA | 0.61 ± 0.05 | 22.7% | 0.38 ± 0.03 | 25.4% | 0.22 ± 0.02 | 24.5% | 0.16 ± 0.02 | 25.1% | 0.16 ± 0.01 | 27.9% |
| BBOA | 3.4 ± 0.41 | 25.2% | 1.2 ± 0.26 | 15.9% | 0.63 ± 0.18 | 13.9% | 0.43 ± 0.14 | 13.4% | 0.21 ± 0.11 | 10.3% |
| LVOOA | 1.04 ± 0.08 | 52.2% | 0.65 ± 0.05 | 58.7% | 0.41 ± 0.04 | 61.5% | 0.29 ± 0.03 | 61.5% | 0.26 ± 0.02 | 61.3% |


Notes: 1) Correlation coefficient (R) for each regression analysis: 0.65 at 370 nm, 0.58 at 470 nm, 0.51 at 520 nm, 0.51 at 570 nm and 0.54 at 660 nm; 2) The
correlation factors for COA and SVOOA are near zero at all wavelength, indicating a negligible contribution from these factors. So only the correlation factors for
HOA, BBOA and LVOOA are listed in the table


Figures

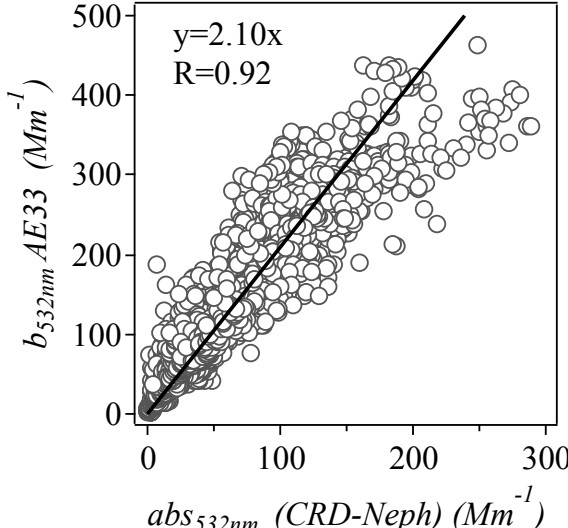


Figure 1. Scatter plot of absorption coefficients at 532 nm measured with aethalometer (AE33)
and those estimated from cavity ring-down spectroscopy (CRD) and Nephelometer measurements.

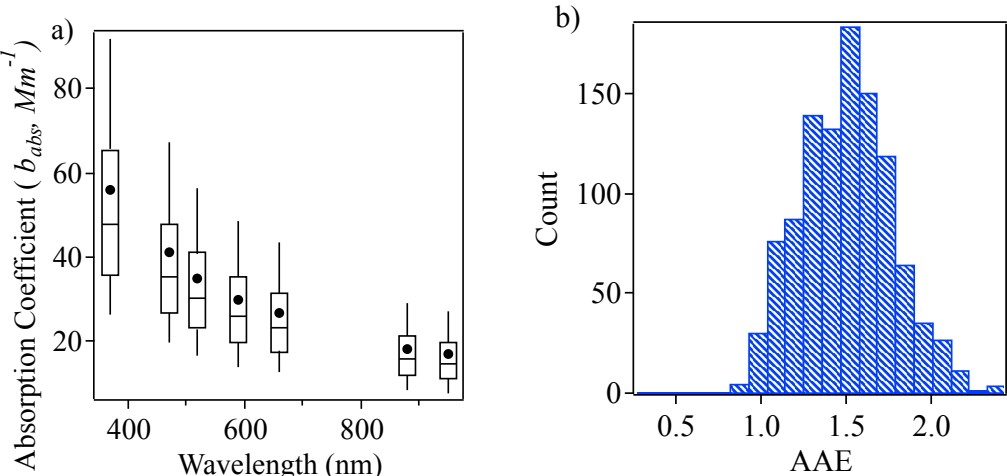


Figure 2. a) Box-whisker plot of absorption coefficient at seven wavelengths as measured with
the AE33; b) Histogram of AAE values over the measurement campaign.

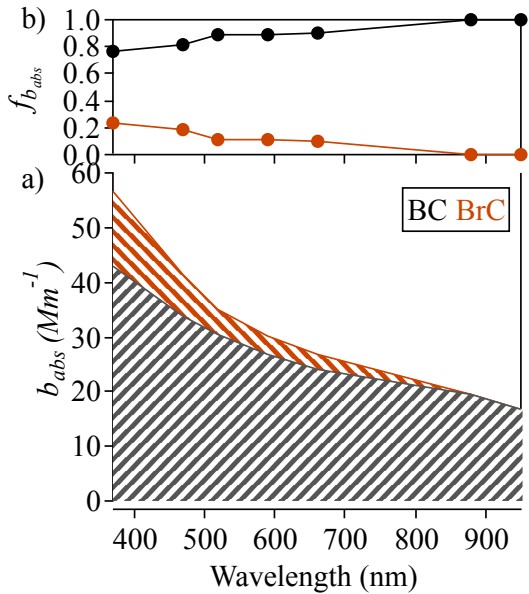


Figure 3. a) Fractions of BC and BrC contributions to aerosol particle light absorption at different
wavelengths; b) Contributions of BC and BrC to the total light absorption coefficient at different
wavelengths.


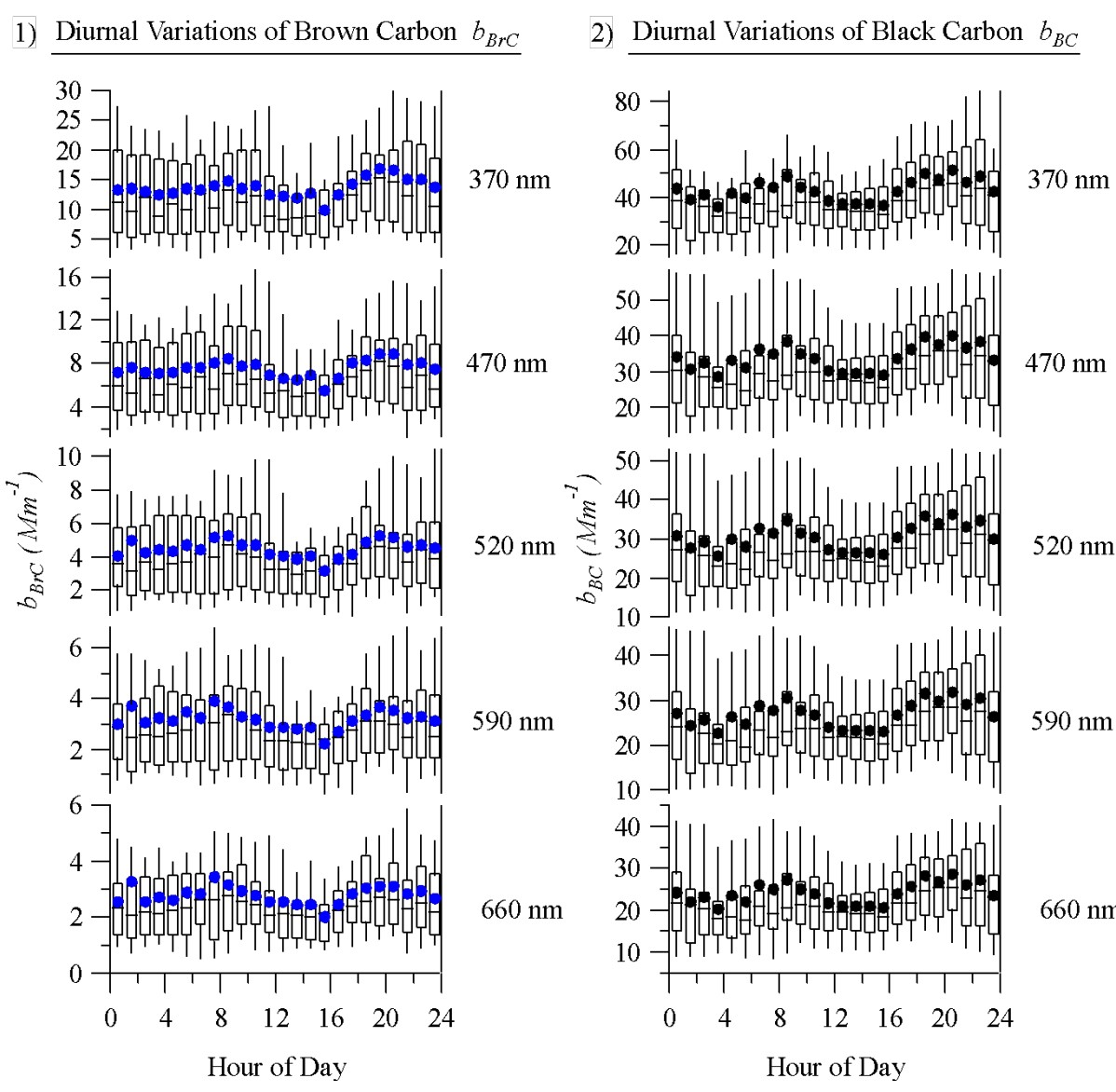


Figure 4. Diurnal variations of BrC and BC light absorption coefficients ($b_{BrC}$ and $b_{BC}$) at different

wavelengths.

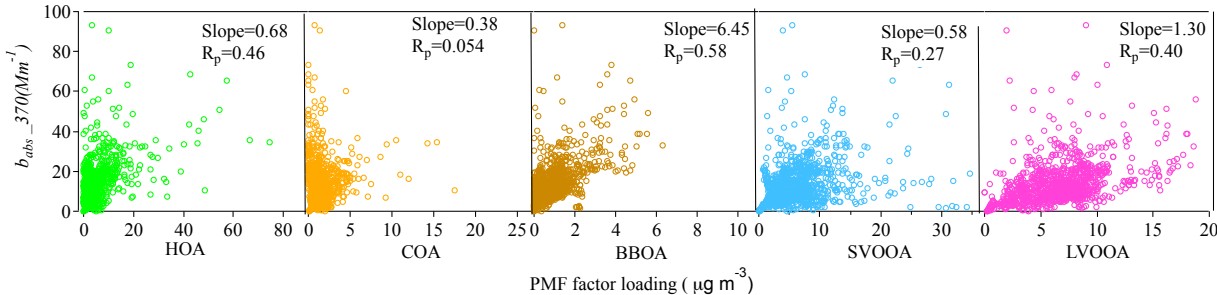

Figure 5. Correlations between the BrC absorption coefficients at 370 nm and the mass loadings
of OA factors resolved by AMS-PMF/ME-2.


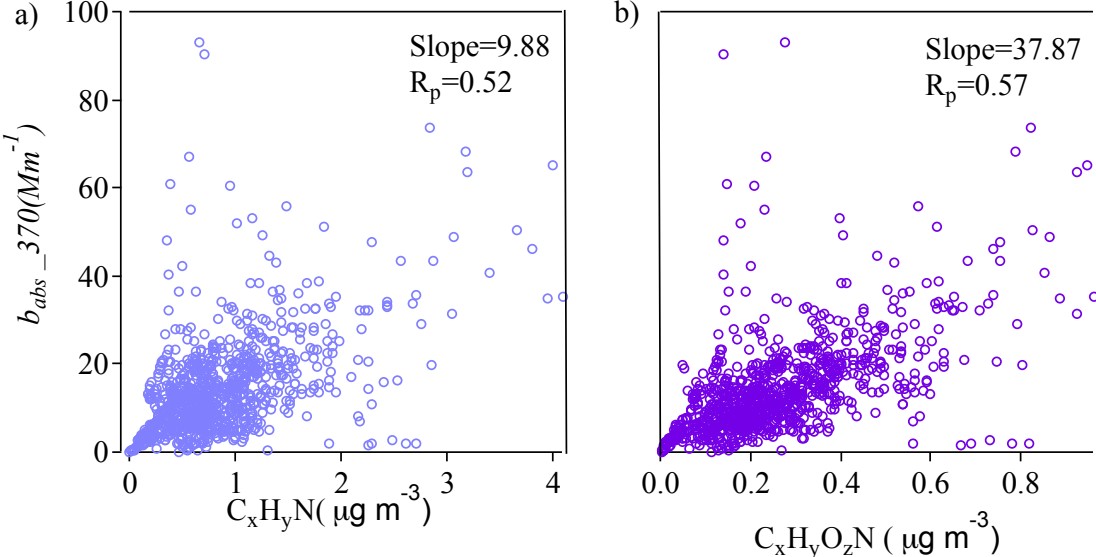


Figure 6. Correlations between BrC absorption coefficients at 370 nm and mass concentrations of N-containing organic ion families.


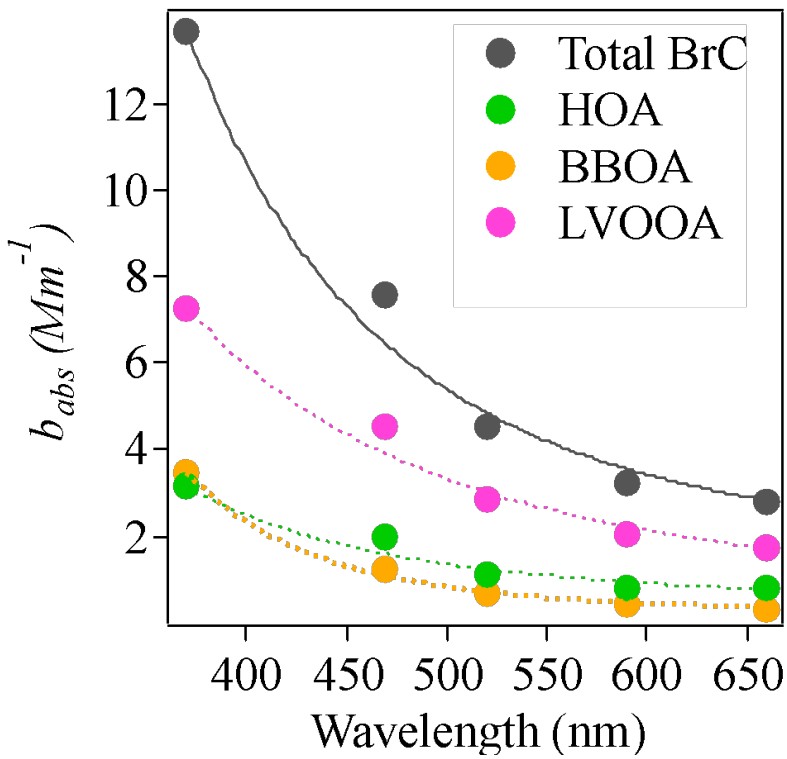


Figure 7. Exponential decay of $b_{abs}$ for total BrC and different light-absorbing OA components
across wavelength.


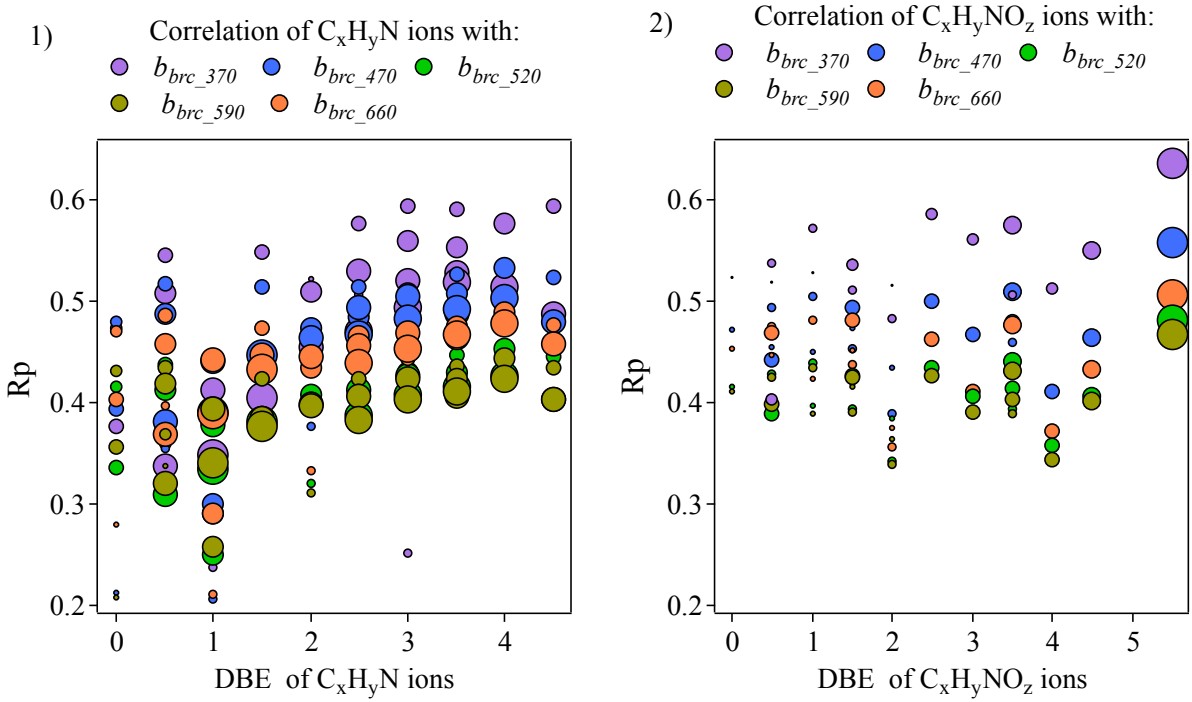


Figure 8. Correlation coefficients between BrC absorption coefficient across different wavelength
and N-containing organic ion fragments grouped by double bond equivalence (panels a and b).
Larger grey dots correspond to higher carbon numbers.