# Peer review of "Chemical characteristics of brown carbon in atmospheric particles at a suburban site near Guangzhou, China"

_Atmospheric Chemistry and Physics, 2018_

## Referee Comment (RC1) · Anonymous Referee #1 · 24 Aug 2018

The manuscript by Qin et al. discusses the possible source of brown carbon at Guanzhou, China. The major finding includes 1) biomass burning is the most important source of brown carbon in the region, and 2) importance of nitrogen containing compounds on brown carbon. The topic will attract the interest of the readers of the journal. The manuscript is clearly written, and easy to understand. I suggest publication of the manuscript after addressing the following comments.

P3L6 'BC is major contributor to light absorption that increases the atmospheric energy budget,'

I am not sure what 'increases' means in this context. Is it possible to make the descrip-

tion to be more specific?

P5 'Measurements and data analysis'

I wonder how the relative humidity of the instruments was controlled, especially for the CRD and nephelometer. As water contents influence both extinction and scattering, it is ideal to have clear descriptions on it. In addition, it is ideal to have comprehensive descriptions on how the instruments were calibrated.

P7L126 'As shown in Figure 2b, the AAE values, which average at 1.43, are almost always higher than 1,'

The histogram has a variation. It would be interesting to discuss how the temporal variation of AAE was controlled.

P8L154 'The diurnal variations of the different wavelengths were not significantly different, although short wavelengths exhibited more obvious diurnal variations. '

There are some interesting patterns in the diurnal variation. For example, the peak at 1AM is clearer for longer wavelength. The morning peak occurred before 8AM for longer wavelength, while it happens after 8AM for shorter wavelength. It would be ideal to have further detailed descriptions on the pattern of the diurnal variation. P9L186 'our results suggest that the absorption coefficient of nascent BBOA' Would you be able to show evidence to consider it as nascent?

P9L202 'light absorption depends on the extent of sp 2 hybridization, in which $\pi$ electrons are usually found'

I wonder what the 'extent of sp2 hybridization' means. Is it the number of sp2 bonding, or length of sp2 conjugated system?

P9L209 'CxHyN + and CxH yOzN ++'

Do they predominantly exist in BBOA, or in other types of OA?

P13L339 'Laskin, J., Laskin, A., Nizkorodov, S. A., Roach, P., Eckert, P., Gilles, M. K., Wang, B., Ji, H., Lee, J. and Hu, Q.: Molecular Selectivity of Brown Carbon Chromophores, 2014.'

Journal name is missing.

---

## Referee Comment (RC2) · Anonymous Referee #2 · 27 Aug 2018

Using a combination of light-absorption measurements (7-wavelength Aethalometer) and chemical-speciation measurements (HR-ToF-AMS) performed in Guangzhou, Qin et al. report 1) contributions of brown carbon to aerosol light absorption, 2) temporal variability of brown-carbon absorption, and 3) correlations between brown-carbon absorption and OA constituents. The manuscript is well written and the topic (sources and speciation of brown carbon) is timely. I believe that this manuscript is suitable for publication in ACP after the following comments are addressed:

General comments:

I believe that the observations could be interrogated further to gain more insight on BrC

sources and optical properties of the various components:

1. How do the diurnal cycles of b_BrC compare to b_BC? This comparison could shed light on how similar the sources of BC and BrC are, and also on the relative contributions of primary versus secondary OA to BrC.

2. As the authors allude to in the Abstract and the Introduction, the light-absorption properties of different BrC species exhibit different wavelength dependence. The data presented in this manuscript could be utilized to further investigate/highlight this. Specifically, I suggest:

a) Extending the analysis in section 2 to present not only MAC values, but also AAE values of the different BrC components.

b) Extending the analysis in section 3 to present the correlations with N-containing ions at longer wavelengths as well, and discuss any differences between different wavelengths.

Specific comments:

Line 6: I see what the authors are trying to say, but the statement that absorption "increases the atmospheric energy budget" is not accurate. The atmosphere does not store energy, but re-emits it back as IR radiation to space. Absorption increases the average temperature of the atmosphere.

Line 7-8: Do you mean 20%-50% of the total aerosol warming (i.e. positive forcing)?

Line 10: Several studies have shown that BrC absorption in the long-visible wavelengths is not negligible (e.g.1–3)

Line 24-27: The authors state that they deal with the effect of coating on AAE in another manuscript, but this should be discussed here as well because it is central to the observations, especially that the average AAE value of 1.43 is at the edge of what has been argued to be just coated BC or BC+BrC.

Line 137-140: The authors reference Lack and Langridge (2013) for the uncertainty in the AAE attribution method, but this is not adequate. The uncertainty should be addressed in this manuscript as well.

References:

(1) Alexander, D. T. L., Crozier, P. a, and Anderson, J. R. (2008) Brown carbon spheres in East Asian outflow and their optical properties. Science 321, 833–6.

(2) Hoffer, A., Tóth, Á., Pósfai, M., Chung, C. E., and Gelencsér, A. (2017) Brown carbon absorption in the red and near-infrared spectral region. Atmos. Meas. Tech. 10, 2353–2359.

(3) Saleh, R., Cheng, Z., and Atwi, K. (2018) The Brown–Black Continuum of Light-Absorbing Combustion Aerosols. Environ. Sci. Technol. Lett. 5, 508–513.

---

## Referee Comment (RC3) · Anonymous Referee #3 · 29 Aug 2018

This manuscript describes measurements of brown carbon and black carbon contributions to aerosol light absorption at a site near Guangzhou, China. The measured brown carbon light absorption is correlated with organic aerosol (OA) composition measured with an AMS. A multiple regression analysis is used to characterize the relationship between brown carbon light absorption and different types of OA species that were obtained via PMF/ME-2 analysis of the AMS data. This manuscript is well written and the work that is described is good and will be of definite interest to the readers of this journal. I recommend publication after the authors address a few minor comments.

1) The authors mention that there is a correlation of brown carbon absorption with

[Figure]

N-containing ions. It would be very useful if the authors could provide a table of the N-containing ions that are observed so that they could be potentially used as tracers and checked for in other sites as well.

2) The authors do not mention how the N-containing ions are distributed across the various OA components. Are they primarily in the BBOA component or are some also found in the LVOOA as well?

3) It would be interesting to see the diurnal cycle in the multiple regression analysis results of scattering at one or more wavelengths.
* * *

---

## Author Comment (AC1) · 29 Oct 2018

**Authors' response to reviewer' comments**

We would like to thank the reviewer for the thoughtful comments and suggestions to improve the manuscript. We address each comment individually below, with the reviewer's comment **in black** and our responses **in blue** and the revised text **in green**.

\*\*\*\*\*\*\*\*\*\*\*\*\*\*\*\*\*\*\*\*\*\*\*\*\*\*\*\*\*\*\*\*\*\*\*\*\*\*\*\*\*\*\*\*\*\*\*\*\*\*\*\*\*\*\*\*\*\*\*\*\*\*\*\*\*\*\*\*\*\*\*\*\*\*\*\*\*\*\*\*\*\*

The manuscript by Qin et al. discusses the possible source of brown carbon at Guanzhou, China. The major finding includes 1) biomass burning is the most important source of brown carbon in the region, and 2) importance of nitrogen containing compounds on brown carbon. The topic will attract the interest of the readers of the journal. The manuscript is clearly written, and easy to understand. I suggest publication of the manuscript after addressing the following comments.
We thank the reviewer for this overall very positive assessment of our manuscript.

1.      P3L6 'BC is major contributor to light absorption that increases the atmospheric energy budget,' I am not sure what 'increases' means in this context. Is it possible to make the description to be more specific?

**Reply:**
We thank the reviewer for pointing out the ambiguous sentence.  We were trying to say a positive radiative forcing. We clarified the sentence in the revised text.

**Revised text:**
BC is major contributor to light absorption that leads to positive radiative forcing, increasing the average temperature of the atmosphere.

2.      P5 'Measurements and data analysis': I wonder how the relative humidity of the instruments was controlled, especially for the CRD and nephelometer. As water contents influence both extinction and scattering, it is ideal to have clear descriptions on it. In addition, it is ideal to have comprehensive descriptions on how the instruments were calibrated.

**Reply:**
The reviewer raised an important point.  For controlling relative humidity (RH), a diffusion drier was used to dry the sampled air stream, which reduced the RH of the air to below 30 %.  The nephelometer was calibrated by $CO_2$ weekly during the field campaign.  Particle-free air was checked once a day. The CRD was calibrated using polystyrene spheres with known indices of refraction before the campaign. We have added the above sentences in the Measurements to make this point clear.

3.      P7L126 'As shown in Figure 2b, the AAE values, which average at 1.43, are almost always higher than 1,' The histogram has a variation. It would be interesting to discuss how the temporal variation of AAE was controlled.

**Reply:**
Thanks for the comment. Total absorption Ångström exponent (AAE) values were calculated by a power-law fitting of the absorption coefficient over all available wavelengths at each time point. Because a plot is generated from the power law fitting at each time point and we have many timepoints, we did not present all the plots in the manuscript. An example of the power law fitting is added in Figure 7 in conjunction with reviewer #2's comment. In terms of the temporal variation, there may be differences in the sources and the relative contribution of each source. We discussed that in section 2 and section 3.

4.      P8L154 'The diurnal variations of the different wavelengths were not significantly different, although short wavelengths exhibited more obvious diurnal variations.' There are some interesting patterns in the diurnal variation. For example, the peak at 1AM is clearer for longer wavelength. The morning peak occurred before 8AM for longer wavelength, while it happens after 8AM for shorter wavelength. It would be ideal to have further detailed descriptions on the pattern of the diurnal variation. P9L186 'our results suggest that the absorption coefficient of nascent BBOA' Would you be able to show evidence to consider it as nascent?

**Reply:**
We agree with the reviewer that the morning peak occurred before 8AM for longer wavelength, while it happened after 8AM for shorter wavelength. However, the peak at 1AM for the long wavelength may be due to some episodic events as the median data is relatively flat. A previous study showed that these changes may be attributed to diurnal changes in BrC sources, which most likely originated from crop residue burning in fall and winter in nearby regions (Wang et al., 2017). We added the discussion as follow in together with Reviewer #2's comment.

**Revised text:**
Figure 4 shows the diurnal variations of both $b_{BrC}$ and $b_{BC}$ at 370, 470, 520, 590, and 660 m respectively. In general, the diurnal cycles of $b_{BrC}$ and $b_{BC}$ share similar patterns, indicating that they may have similar sources. However, it should be noted that some OA factors, such as BBOA and HOA, also share similar pattern(Qin et al., 2017) .Overall, there were two peaks at each wavelength. The first peak appeared in the morning at around 8:00 LT, with a peak before 8:00 LT for longer wavelength and after 8:00 LT for shorter wavelength. The second peak appeared at 21:00 LT and its intensity decreased untill 24:00 LT. These changes may be attributed to diurnal changes in sources, which most likely originated from crop residue burning in fall and winter in nearby regions(Wang et al., 2017).

[Figure]

1) Diurnal Variations of Brown Carbon $b_{BrC}$

2) Diurnal Variations of Black Carbon $b_{BC}$

5. P9L202 'light absorption depends on the extent of sp 2 hybridization, in which electrons are usually found. I wonder what the 'extent of sp2 hybridization' means. Is it the number of sp2 bonding, or length of sp2 conjugated system?

**Reply:**
Thanks for the comments. We have classified this point in the revised text. By 'extent of sp2 hybridization' we mean the length of the conjugated system. As the conjugation gets larger, the energy difference between the excited state and the ground state goes down, which makes the absorption band shift to longer wavelengths.

6.        P9L209 'CxHyN + and CxH yOzN ++'Do they predominantly exist in BBOA, or in other types of OA?

**Reply:**
The reviewer raised an important point. As shown in the following figure, the N-containing ion fragments are distributed in all the OA factors, although the relative contribution is higher in BBOA than that in other OA factors.  However, as the signal intensities are already normalized in the PMF analysis, the distribution of these fragments among the OA factors also depend on the mass concentration of each OA factor. We have added this discussion in the supporting information.

[Figure]

7.        P13L339 'Laskin, J., Laskin, A., Nizkorodov, S. A., Roach, P., Eckert, P., Gilles, M. K., Wang, B., Ji, H., Lee, J. and Hu, Q.: Molecular Selectivity of Brown Carbon Chromophores, 2014.'Journal name is missing.

**Response:**
Added as suggested.

Reference:

Qin, Y. M., Tan, H. B., Li, Y. J., Schurman, M. I., Li, F., Canonaco, F., Prévôt, A. S. H. and Chan, C. K.: Impacts of traffic emissions on atmospheric particulate nitrate and organics at a downwind site on the periphery of Guangzhou, China, Atmos. Chem. Phys., 2017(x), 1–31, doi:10.5194/acp-2017-116, 2017.

Wang, Y., Hu, M., Lin, P., Guo, Q., Wu, Z., Li, M., Zeng, L., Song, Y., Zeng, L., Wu, Y., Guo, S., Huang, X. and He, L.: Molecular Characterization of Nitrogen-Containing Organic Compounds in Humic-like Substances Emitted from Straw Residue Burning, Environ. Sci. Technol., 51(11), 5951–5961, doi:10.1021/acs.est.7b00248, 2017.

---

## Author Comment (AC2) · 29 Oct 2018

**Authors' response to reviewer' comments**

We would like to thank the reviewer for the thoughtful comments and suggestions to improve the manuscript. We address each comment individually below, with the reviewer' comment **in black** and our responses **in blue** and the revised text **in green**.
* * *
**Response to Reviewer #2:**

Using a combination of light-absorption measurements (7-wavelength Aethalometer) and chemical-speciation measurements (HR-ToF-AMS) performed in Guangzhou, Qin et al. report 1) contributions of brown carbon to aerosol light absorption, 2) temporal variability of brown-carbon absorption, and 3) correlations between brown-carbon absorption and OA constituents. The manuscript is well written and the topic (sources and speciation of brown carbon) is timely. I believe that this manuscript is suitable for publication in ACP after the following comments are addressed:

General comments:
I believe that the observations could be interrogated further to gain more insight on BrC sources and optical properties of the various components:

1.      How do the diurnal cycles of b_BrC compare to b_BC? This comparison could shed light on how similar the sources of BC and BrC are, and also on the relative contributions of primary versus secondary OA to BrC.

**Reply:**
The reviewer raised an important point. We have replaced the original Figure 4 showing the diurnal cycles of $b_{BrC}$ to a revised Figure 4 showing the diurnal cycles of $b_{BrC}$ and $b_{BC}$.

**Revised text:**
Figure 4 shows the diurnal variations of both $b_{BrC}$ and $b_{BC}$ at 370, 470, 520, 590, and 660 m respectively. In general, the diurnal cycles of $b_{BrC}$ and $b_{BC}$ share similar patterns, indicating that they may have similar sources. However, it should be noted that some OA factors, such as BBOA and HOA, also share similar patterns (Qin et al., 2017). Overall, there were two peaks at each wavelength. The first peak appeared in the morning at around 8:00 LT, with a peak before 8:00 LT for longer wavelength and after 8:00 LT for shorter wavelength. The second peak appeared at 21:00 LT and its intensity decreased until 24:00 LT. These changes may be attributed to diurnal changes in particle source, which most likely originated from crop residue burning in the fall and winter in nearby regions (Wang et al., 2017)

[Figure]

1) Diurnal Variations of Brown Carbon $b_{BrC}$

2) Diurnal Variations of Black Carbon $b_{BC}$

Hour of Day

Hour of Day

2. As the authors allude to in the Abstract and the Introduction, the light-absorption properties of different BrC species exhibit different wavelength dependence. The data presented in this manuscript could be utilized to further investigate/highlight this.

**Reply:**
We thank the reviewer for this thoughtful suggestion. Below is the response to each suggestion.

Specifically, I suggest:
Extending the analysis in section 2 to present not only MAC values, but also AAE values of the different BrC components.

**Reply:**

The figure below shows the exponential decay of $b_{abs}$ for different light-absorbing components. The fitted AAE values for those components are 3.52, 3.28, 5.50 and 2.67 for total BrC, HOA, BBOA and LVOOA respectively. These results indicate that variability of AAE values ranging from different sources which is likely inherent to the chemical variability of BrC constituents. We have now included them in Figure 7 in main text and discuss this point in Line 220-224 on Page 9.

[Figure]

Extending the analysis in section 3 to present the correlations with N-containing ions at longer wavelengths as well, and discuss any differences between different wavelengths.

**Reply:**

Figure below shows more correlation analysis between $b_{Brc}$ at different wavelength and DBE of $C_xH_yN$ and $C_xH_yNO_z$ ions. The Pearson's R ($R_p$) values are in general consistent with what we have shown in Figure 8 in the original main text.

[Figure]

**Revised text:**

Figure 8 shows the correlation coefficients between $b_{BrC}$ at all available wavelengths and the mass loadings of each ion in $C_xH_yN^+$ and $C_xH_yNO_z^+$ families at different DBE values. For the $C_xH_yN^+$ family, $R_p$ increased as DBE increased across all wavelength, suggesting that $b_{BrC}$ was better correlated with fragments with higher degrees of unsaturation or cyclization. And increasing trend of $R_p$ as DBE increased is more obvious for short wavelengths (e.g. $\lambda$ at 370 nm and 470 nm), suggesting that the absorption at short wavelengths are more associated with the unsaturation or cyclization. Indeed, in saturated organics, light absorption involves excitation of n electrons, which requires more energy and, therefore, shorter incident wavelengths (e.g., short UV). In unsaturated organics, the delocalized $\pi$ electrons are in clusters of sp2 hybrid bonds and in longer conjugated systems, such that the energy difference between the excited state and the ground state goes down, which makes the absorption band shift to longer wavelengths. These structural features may explain in part the increased correlation between mass loadings of the $C_xH_yN^+$ family and light absorption with decreasing ion saturation. For the $C_xH_yO_zN^+$ family, we did not observe obvious trends in the correlation coefficient with changing degree of saturation/cyclization (Figure 8b). This phenomenon is consistent across different wavelength. However, the overall Pearson's Rs of $b_{BrC}$ with $C_xH_yO_zN^+$ were higher than those with $C_xH_yN^+$. The $R_p$ for each group of ions is higher at short wavelengths ($\lambda$ at 370 nm and 470 nm).

Specific comments:

3.    Line 6: I see what the authors are trying to say, but the statement that absorption "increases the atmospheric energy budget" is not accurate. The atmosphere does not store energy, but re-emits it back as IR radiation to space. Absorption increases the average temperature of the atmosphere.

**Reply:**

We thank the reviewer to point out the ambiguous sentence. We clarified the sentence in the revised text.

**Revised text:**
BC is a major contributor to light absorption that leads to positive radiative forcing, increasing the average temperature of the atmosphere.

4.      Line 7-8: Do you mean 20%-50% of the total aerosol warming (i.e. positive forcing)?

**Reply:**
Thanks for pointing out the ambiguous sentence. We meant the 20%-50% of total aerosol light absorption.  We have revised the sentence as follow:

**Revised text:**
The BrC absorption contribution to total aerosol light absorption can reach 20–50% over regions dominated by seasonal biomass burning and biofuel combustion (Feng et al., 2013).

5.      Line 10: Several studies have shown that BrC absorption in the long-visible wavelengths is not negligible (e.g.1–3)

 **Reply:**
Thanks for pointing out the misleading sentence. Yes, we agree that BrC absorption in the long-visible wavelengths is not negligible. We were trying to distinguish the absorption properties of BrC and BC which makes the AAE attribution method possible. A revised text have been added.

**Revised text:**
A significant difference in optical feature of  BrC and BC is that BrC absorbs light primarily at UV and short-visible wavelengths with the absorption decreasing significantly at long wavelengths, while BC absorbs strongly and constantly throughout the UV to visible spectrum (Andreae and Gelencsér, 2006; Bergstrom et al., 2007; Bond and Bergstrom, 2006).

6.      Line 24-27: The authors state that they deal with the effect of coating on AAE in another manuscript, but this should be discussed here as well because it is central to the observations, especially that the average AAE value of 1.43 is at the edge of what has been argued to be just coated BC or BC+BrC.

**Reply:**
The reviewer raised an important point. A Mie theory model was used to estimate the AAE for BC-containing particles ($AAE_{BC}$) at core-shell scenarios with different refractive indexes. A detailed discussion is presented in another manuscript. Briefly, $AAE_{BC}$ is sensitive to specific refractive index of core and shell of the particles and the size of the particle. The size distribution is from scanning mobility particle sizer and aerodynamic particle sizer measurement, and we vary the refractive index of the core and shell in the model. The method is adopted from a previous publication from the group (Tan et al., 2016). In general, $AAE_{BC}$ increases as the real part refractive index of the core increases or the imaginary decreases, or alternatively real part of the shell

increases. The $AAE_{BC}$ ranges from 0.67-1.03 across the different scenario. Table S1 is added in the revised manuscript.

**Table S1. $AAE_{BC}$ estimation from Mie theory model**

| Model run number | Refractive index | | | | AAE |
| :---: | :---: | :---: | :---: | :---: | :---: |
| | Core | | Shell | | |
| | Real part | Imaginary part | Real part | Imaginary part | |
| 1 | 1.6 | 0.54i | 1.55 | 0.0000001i | 0.848518188 |
| 2 | 1.7 | 0.54i | 1.55 | 0.0000001i | 0.871846684 |
| 3 | 1.8 | 0.54i | 1.55 | 0.0000001i | 0.89561921 |
| 4 | 1.9 | 0.54i | 1.55 | 0.0000001i | 0.919776955 |
| 5 | 2 | 0.54i | 1.55 | 0.0000001i | 0.943934591 |
| 6 | 1.8 | 0.4i | 1.55 | 0.0000001i | 0.979578577 |
| 7 | 1.8 | 0.5i | 1.55 | 0.0000001i | 0.91879886 |
| 8 | 1.8 | 0.6i | 1.55 | 0.0000001i | 0.862171196 |
| 9 | 1.8 | 0.7i | 1.55 | 0.0000001i | 0.809566808 |
| 10 | 1.8 | 0.8i | 1.55 | 0.0000001i | 0.760456075 |
| 11 | 1.8 | 0.9i | 1.55 | 0.0000001i | 0.714608394 |
| 12 | 1.8 | 1.0i | 1.55 | 0.0000001i | 0.671630187 |
| 13 | 1.8 | 0.54i | 1.35 | 0.0000001i | 0.885192669 |
| 14 | 1.8 | 0.54i | 1.4 | 0.0000001i | 0.887286337 |
| 15 | 1.8 | 0.54i | 1.45 | 0.0000001i | 0.8885085 |
| 16 | 1.8 | 0.54i | 1.5 | 0.0000001i | 0.890599011 |
| 17 | 1.8 | 0.54i | 1.55 | 0.0000001i | 0.89561921 |
| 18 | 1.8 | 0.54i | 1.6 | 0.0000001i | 0.905391588 |
| 19 | 2 | 0.4i | 1.6 | 0.0000001i | 1.035139318 |

7.      Line 137-140: The authors reference Lack and Langridge (2013) for the uncertainty in the AAE attribution method, but this is not adequate. The uncertainty should be addressed in this manuscript as well.

**Reply:**

Uncertainty of the BrC light absorption from the AAE attribution method is primarily from uncertainty of choice of $AAE_{BC}$. Sensitivity analysis of BrC contribution to total light absorption is added based on the $AAE_{BC}$ from Mie theory model output. We have added the following discussion in the revised manuscript in main text Line 171-173 and Figure S1.

**Revised text:**

[Figure]

Reference:
Andreae, M. O. and Gelencsér, A.: Black carbon or brown carbon? The nature of light-absorbing

carbonaceous aerosols, Atmos. Chem. Phys., 6(3), 3419–3463, doi:10.5194/acpd-6-3419-2006, 2006.

Bergstrom, R. W., Pilewskie, P., Russell, P. B., Redemann, J., Bond, T. C., Quinn, P. K. and Sierau, B.: Spectral absorption properties of atmospheric aerosols, Atmos. Chem. Phys., 7(23), 5937–5943, doi:10.5194/acp-7-5937-2007, 2007.

Bond, T. C. and Bergstrom, R. W.: Light Absorption by Carbonaceous Particles: An Investigative Review, Aerosol Sci. Technol., 40(1), 27–67, doi:10.1080/02786820500421521, 2006.

Feng, Y., Ramanathan, V. and Kotamarthi, V. R.: Brown carbon: A significant atmospheric absorber of solar radiation, Atmos. Chem. Phys., 13(17), 8607–8621, doi:10.5194/acp-13-8607-2013, 2013.

Qin, Y. M., Tan, H. B., Li, Y. J., Schurman, M. I., Li, F., Canonaco, F., Prévôt, A. S. H. and Chan, C. K.: Impacts of traffic emissions on atmospheric particulate nitrate and organics at a downwind site on the periphery of Guangzhou, China, Atmos. Chem. Phys., 2017(x), 1–31, doi:10.5194/acp-2017-116, 2017.

Tan, H., Liu, L., Fan, S., Li, F., Yin, Y., Cai, M. and Chan, P. W.: Aerosol optical properties and mixing state of black carbon in the Pearl River Delta, China, Atmos. Environ., 131, 196–208, doi:10.1016/j.atmosenv.2016.02.003, 2016.

Wang, Y., Hu, M., Lin, P., Guo, Q., Wu, Z., Li, M., Zeng, L., Song, Y., Zeng, L., Wu, Y., Guo, S., Huang, X. and He, L.: Molecular Characterization of Nitrogen-Containing Organic Compounds in Humic-like Substances Emitted from Straw Residue Burning, Environ. Sci. Technol., 51(11), 5951–5961, doi:10.1021/acs.est.7b00248, 2017.

---

## Author Comment (AC3) · 29 Oct 2018

**Authors' Response to Reviewer' Comments**

We would like to thank the reviewer for the thoughtful comments and suggestions to improve the manuscript. We address each comment individually below, with the reviewer' comment **in black** and our responses **in blue** and the revised text **in green**.

\*\*\*\*\*\*\*\*\*\*\*\*\*\*\*\*\*\*\*\*\*\*\*\*\*\*\*\*\*\*\*\*\*\*\*\*\*\*\*\*\*\*\*\*\*\*\*\*\*\*\*\*\*\*\*\*\*\*\*\*\*\*\*\*\*\*\*\*\*\*\*\*\*\*\*\*\*\*\*\*\*\*\*\*\*\*\*\*\*\*\*\*

 **Response to Reviewer #3:**

This manuscript describes measurements of brown carbon and black carbon contributions to aerosol light absorption at a site near Guangzhou, China. The measured brown carbon light absorption is correlated with organic aerosol (OA) composition measured with an AMS. A multiple regression analysis is used to characterize the relationship between brown carbon light absorption and different types of OA species that were obtained via PMF/ME-2 analysis of the AMS data. This manuscript is well written and the work that is described is good and will be of definite interest to the readers of this journal. I recommend publication after the authors address a few minor comments.

We thank the reviewer for the suggestions. Below is the response to each suggestion.

1.     The authors mention that there is a correlation of brown carbon absorption with N-containing ions. It would be very useful if the authors could provide a table of the N-containing ions that are observed so that they could be potentially used as tracers and checked for in other sites as well.

**Reply:**
Table S2 below shows the N-containing ions and their Pearson's correlation coefficients ($R_p$) between absorption at each wavelength and the DBE of each ion. This table has been added to SI as Table S2.

**Revised text:**
**Table S2:** N-containing ions and their respective DBE and Rp respective with each wavelength

| Ions | DBE | Rp_370 | Rp_470 | Rp_520 | Rp_590 | Rp_660 |
|------|-----|--------|--------|--------|--------|--------|
| CHN | 2 | 0.52 | 0.46 | 0.40 | 0.40 | 0.44 |
| CH4N | 0.5 | 0.34 | 0.37 | 0.32 | 0.34 | 0.41 |
| CH5N | 0 | 0.18 | 0.21 | 0.17 | 0.21 | 0.28 |
| C2HN | 3 | 0.25 | 0.18 | 0.15 | 0.13 | 0.12 |
| C2H2N | 2.5 | 0.51 | 0.45 | 0.38 | 0.38 | 0.43 |
| C2H3N | 2 | 0.46 | 0.38 | 0.32 | 0.31 | 0.33 |

| | | | | | |
|---|---|---|---|---|---|
| C2H4N | 1.5 | 0.44 | 0.44 | 0.37 | 0.38 | 0.45 |
| C2H5N | 1 | 0.24 | 0.21 | 0.17 | 0.18 | 0.21 |
| C2H6N | 0.5 | 0.32 | 0.35 | 0.30 | 0.33 | 0.40 |
| C3H7N | 1 | 0.09 | 0.06 | 0.05 | 0.05 | 0.06 |
| C3H8N | 0.5 | 0.36 | 0.42 | 0.36 | 0.37 | 0.43 |
| C3H9N | 0 | 0.47 | 0.48 | 0.42 | 0.43 | 0.47 |
| C4H2N | 4.5 | 0.59 | 0.52 | 0.45 | 0.43 | 0.48 |
| C4H4N | 3.5 | 0.59 | 0.53 | 0.45 | 0.44 | 0.48 |
| C4H5N | 3 | 0.59 | 0.51 | 0.43 | 0.42 | 0.44 |
| C4H6N | 2.5 | 0.58 | 0.51 | 0.43 | 0.42 | 0.47 |
| C4H8N | 1.5 | 0.55 | 0.51 | 0.43 | 0.42 | 0.47 |
| C4H9N | 1 | 0.06 | 0.08 | 0.06 | 0.06 | 0.07 |
| C4H10N | 0.5 | 0.54 | 0.52 | 0.44 | 0.43 | 0.49 |
| C4H11N | 0 | 0.38 | 0.39 | 0.34 | 0.36 | 0.40 |
| C5H5N | 4 | 0.58 | 0.53 | 0.45 | 0.44 | 0.49 |
| C5H6N | 3.5 | 0.55 | 0.51 | 0.43 | 0.42 | 0.48 |
| C5H7N | 3 | 0.56 | 0.51 | 0.43 | 0.42 | 0.47 |
| C5H8N | 2.5 | 0.49 | 0.48 | 0.41 | 0.41 | 0.47 |
| C5H9N | 2 | 0.51 | 0.47 | 0.41 | 0.40 | 0.43 |
| C5H11N | 1 | 0.29 | 0.30 | 0.25 | 0.26 | 0.29 |
| C5H12N | 0.5 | 0.51 | 0.49 | 0.41 | 0.42 | 0.46 |
| C6H6N | 4.5 | 0.49 | 0.48 | 0.40 | 0.40 | 0.46 |
| C6H8N | 3.5 | 0.53 | 0.49 | 0.41 | 0.41 | 0.46 |
| C6H9N | 3 | 0.52 | 0.50 | 0.43 | 0.42 | 0.47 |
| C6H10N | 2.5 | 0.53 | 0.49 | 0.41 | 0.41 | 0.46 |

| | | | | | |
|---|---|---|---|---|---|
| C6H11N | 2 | 0.45 | 0.46 | 0.40 | 0.40 | 0.45 |
| C6H12N | 1.5 | 0.44 | 0.44 | 0.38 | 0.38 | 0.45 |
| C6H13N | 1 | 0.41 | 0.44 | 0.38 | 0.39 | 0.44 |
| C6H14N | 0.5 | 0.34 | 0.38 | 0.31 | 0.32 | 0.37 |
| C7H9N | 4 | 0.51 | 0.50 | 0.43 | 0.42 | 0.48 |
| C7H10N | 3.5 | 0.52 | 0.49 | 0.42 | 0.41 | 0.47 |
| C7H11N | 3 | 0.49 | 0.48 | 0.41 | 0.40 | 0.45 |
| C7H12N | 2.5 | 0.47 | 0.47 | 0.39 | 0.38 | 0.44 |
| C8H14N | 1.5 | 0.40 | 0.45 | 0.38 | 0.38 | 0.43 |
| C8H15N | 1 | 0.35 | 0.39 | 0.33 | 0.34 | 0.39 |
| CHNO | 2 | 0.52 | 0.43 | 0.38 | 0.36 | 0.37 |
| CH2NO | 1.5 | 0.54 | 0.47 | 0.42 | 0.42 | 0.45 |
| CH3NO | 1 | 0.53 | 0.45 | 0.40 | 0.39 | 0.42 |
| CH4NO | 0.5 | 0.52 | 0.45 | 0.40 | 0.41 | 0.45 |
| CH5NO | 0 | 0.52 | 0.47 | 0.42 | 0.41 | 0.45 |
| C2HNO | 3 | 0.17 | 0.08 | 0.07 | 0.04 | 0.01 |
| C2H2NO | 3.5 | 0.51 | 0.46 | 0.39 | 0.39 | 0.43 |
| C2H3NO | 2 | 0.48 | 0.39 | 0.34 | 0.34 | 0.36 |
| C2H4NO | 1.5 | 0.51 | 0.45 | 0.39 | 0.39 | 0.44 |
| C2H5NO | 1 | 0.57 | 0.50 | 0.44 | 0.43 | 0.48 |
| C2H6NO | 0.5 | 0.54 | 0.49 | 0.43 | 0.42 | 0.47 |
| C3HNO | 4 | 0.51 | 0.41 | 0.36 | 0.34 | 0.37 |
| C3H3NO | 3 | 0.56 | 0.47 | 0.41 | 0.39 | 0.41 |
| C3H2NO | 3.5 | 0.58 | 0.48 | 0.41 | 0.40 | 0.43 |
| C3H4NO | 2.5 | 0.59 | 0.50 | 0.43 | 0.43 | 0.46 |

| | | | | | | |
|---|---|---|---|---|---|---|
| C3H6NO | 1.5 | 0.54 | 0.49 | 0.43 | 0.42 | 0.48 |
| C3H8NO | 0.5 | 0.40 | 0.44 | 0.39 | 0.40 | 0.47 |
| C4H2NO | 4.5 | 0.55 | 0.46 | 0.41 | 0.40 | 0.43 |
| C4H4NO | 3.5 | 0.58 | 0.51 | 0.44 | 0.43 | 0.48 |
| C7H6NO3 | 5.5 | 0.64 | 0.56 | 0.48 | 0.47 | 0.51 |

2.      The authors do not mention how the N-containing ions are distributed across the various OA components. Are they primarily in the BBOA component or are some also found in the LVOOA as well?

**Reply:**
The reviewer raised an important point. They are distributed within all the OA factors, while the relative contribution is higher in BBOA.  However, as in the signal intensities are already normalized in the PMF analysis, the distribution of these fragments among the OA factors also depends on the mass concentration of each OA factor. This information is shown below in Figure S2 and added to SI as Figure S2.

[Figure]

3.     It would be interesting to see the diurnal cycle in the multiple regression analysis results of scattering at one or more wavelengths.

**Reply:**

We thank reviewer for the suggestion. We agree that the diurnal cycle will provide very insightful information.  However, the information with diurnal cycle from multiple regression analysis can be reflected from the loading contribution from each OA factor. The mass absorption coefficient (MAC) for each OA factor is constant across time. In the multiple linear regression, the diurnal variations across different wavelengths bear the same feature with mass loadings of OA factors. The reason is that absorption coefficients are tightly related to the mass concentration of each OA sources (absorption coefficients=MAC*mass concentration).  Nevertheless, this information is shown below in Figure S3 and added to SI as Figure S3.

**Revised text:**
The diurnal variation of absorption coefficients from each OA component and its relative contribution to absorption at 370 nm is shown as follows. Overall, there was no obvious diurnal variation for the absorption coefficients of LVOOA, while there were obvious nighttime and rush hour increases for HOA. The absorption coefficients of BBOA also slightly increased during nighttime and decreased in the mid-day. As these absorption coefficients are tightly related to the

mass concentration of each OA source, they shared exactly the same diurnal pattern as the mass concentration of each OA factors.